# Seismic Response Effect on Base-Isolated Rigid Structures by Mass Eccentricity in Nuclear Plants

Tae-Myung Shin [1,*] and Byung-Chan Lee [2]

1 Aeronautical & Mechanical Design Engineering, Korea Nat'l University of Transportation, Cheongju 27469, Republic of Korea
2 Environmental Engineering, Korea Nat'l University of Transportation, Cheongju 27469, Republic of Korea; bclee@ut.ac.kr
* Correspondence: tmshin@ut.ac.kr

**Featured Application: The results can be applicable to the plant equipment design for seismic enhancement.**

**Abstract:** The purpose of this paper is to analyze the seismic response effect caused by the mass eccentricity of individual equipment when conducting base isolation for the improvement of the seismic performance of a nuclear power plant. Recent research has interpreted and confirmed through analysis and testing that base isolation for safety-related equipment in nuclear power plants is an efficient alternative to designing for excessive seismic loads. Depending on the equipment, unavoidable mass eccentricity can occur, which necessitates verification of the response impact caused by eccentricity. In this paper, we analyze the seismic response impact of equipment with mass eccentricity using small base isolators. To do so, sensitivity analysis of the seismic response due to mass eccentricity is conducted for a base-isolated concentrated mass model. Furthermore, three efficient mass eccentricity models suitable for testing are designed and manufactured. Simulation analyses using the finite element method (FEM) models are performed, followed by three-axis shake table tests to validate the seismic response impact of mass eccentricity. In conclusion, it is confirmed that applying small base isolators to equipment with mass eccentricity can affect seismic response impact to some extent when compared for a beyond-design-basis earthquake (BDBE).

**Keywords:** mass eccentricity; seismic response; base isolation; beyond-design-basis earthquake; LRB; nuclear power plant; equipment; seismic analysis; sensitivity analysis

## 1. Introduction

The recent earthquakes in Gyeongju and Pohang have been of the largest magnitude among recorded earthquakes in South Korea, raising significant national concerns due to their considerable impact. Furthermore, concerns about the safety of nuclear power plants located near earthquake-prone areas have led to the development of technologies to enhance the seismic performance of these nuclear facilities. Within nuclear power plants, there are numerous large and small pieces of equipment and structures. Typically, during the design phase, efforts are made to avoid eccentricity in mass or stiffness as much as possible. However, in some cases, unavoidable structural characteristics or modifications during operation, such as the installation of additional equipment, can result in a shift in the center of mass (or stiffness). For example, equipment such as emergency diesel generators and spent fuel storage racks within nuclear power plants may unavoidably exhibit significant mass eccentricities due to structural arrangements dictated by their design functions. When the mass center is offset, the natural frequencies or mode characteristics of the structures can change, ultimately leading to a potential increase in seismic response. There have been many studies introducing the seismic response effect and warning about response increases in building structures due to mass or stiffness eccentricity. However, there are

very few studies on the effect of mass eccentricity on the seismic response of base-isolated equipment in building structures or operating plants.

In recent years, it has been proposed to apply seismic isolation technology to major individual facilities with high safety importance to improve the seismic performance of operating nuclear power plants (NPPs). The application of small laminated rubber bearing (LRB) can be a good option for the purpose. Therefore, this paper aims to investigate the influence of mass eccentricity on the seismic response of base-isolated equipment. To achieve this, sensitivity analysis (SA) of the seismic response to mass eccentricity was performed to analyze its impact on the maximum acceleration (and displacement) response during design-basis earthquakes. Some specific eccentricity models conducive to experimentation were set up, and simulation analyses using the test models were conducted to predict the test results. Subsequently, shake table tests were performed to compare and validate these analytical results.

## 2. Analysis of Seismic Response in Base-Isolated Mass-Eccentric Structures

### 2.1. Response Characteristics Due to Eccentricity

When examining examples of research related to the seismic response characteristics of structures due to eccentricity, it has been reported that, when conducting three-dimensional structural analysis on high-rise buildings, when the center of gravity in the horizontal direction is eccentric, torsional effects occur, along with bending, leading to reduced lateral load-bearing capacity [1]. This becomes more distinct if the structure is irregular along the height, according to some studies [2,3]. Furthermore, the seismic response impact on structures with eccentricity has been observed to increase with larger eccentricities. Although maximum displacements decrease, increased torsional behavior results in concentrated damage, leading to greater damage [4].

In the case of base-isolated structures, the response impact due to the distance between the mass center and the stiffness center of the base isolation layer has been examined. Research indicates that, while maximum response displacement does not vary significantly, maximum response acceleration, inter-story shear forces, and forces on upper floors significantly increase with larger eccentricities [5,6]. Excessive damage is expected in some components as a result. Therefore, it is crucial to avoid eccentricity in structural placement, and it is recommended to design the mass center and stiffness center of the building to be as close as possible within allowable limits [7,8].

### 2.2. Analysis of Design Requirements for Mass Eccentricity

When referring to the Building Seismic Design Standards, KDS 41-17, provided by the Ministry of Land, Infrastructure, and Transport [9], if the maximum floor displacement, considering accidental eccentricity in one direction, is greater than 1.2 times the average floor displacement at both ends of a structure orthogonal to that direction, it is considered to be a torsionally irregular structure. Therefore, for torsionally irregular buildings, the torsional amplification factor $A_x$ should be multiplied by the accidental torsional moment at each floor for calculation purposes.

$$A_x = \left( \frac{\delta_{max}}{1.2\delta_{avg}} \right)^2 \tag{1}$$

($\delta_{max}$: Maximum displacement at *x*-th floor level, $\delta_{avg}$: Average displacement at each corner of *x*-th floor level).

In addition, according to the Safety Review Guide for Pressurized Water Reactors provided by the Korea Institute of Nuclear Safety (KINS) [10], structures, systems, and components classified as Seismic Category I should consider torsional effects during seismic analysis. In particular, the possibility of accidental torsion should also be considered. If the influence of eccentric mass on torsion during seismic analysis is deemed significant, the eccentric mass and its eccentricity should be included in the analysis model.

Japan's JNES 2013 [11], concerning design policies for base-isolated structures, suggests that, for base-isolated structures, when the distance between the stiffness center and the mass center is significant, torsional behavior during an earthquake may occur, resulting in increased relative displacement. This consideration is due to the potential for eccentricity to cause structural damage through increased stress and potential collisions with nearby structures. Therefore, during seismic design, efforts should be made to place the stiffness center and mass center as close as possible. If there is a significant distance between the stiffness and mass centers, dynamic behavior characteristics and their impact should be assessed through analysis or testing, and appropriate measures should be taken.

Furthermore, the American Society of Civil Engineers (ASCE) 4-16 [12], which provides seismic analysis guidelines for safety-related nuclear structures, requires the appropriate representation of the structure's mass center and stiffness center positions to explain the torsional effect caused by eccentricity. Therefore, in the case of structures where the mass center and stiffness center do not coincide, measures should be taken to ensure that the torsional stiffness and mass moment of inertia are considered in light of eccentricity. In summary, both domestic and international seismic design requirements emphasize the importance of conducting detailed analyses to understand dynamic characteristics and assess the impact of eccentricity on structural response when structures have significant levels of stiffness or mass eccentricity. Taking necessary actions in the design phase based on these assessments is crucial.

## 3. Sensitivity Analysis of Base Isolation Response Due to Mass Eccentricity

### 3.1. Design of Small Base Isolators for Nuclear Power Plant Equipment

As a measure to enhance the seismic performance of internal equipment and facilities in operational nuclear power plants during beyond-design-basis earthquakes (BDBE) exceeding the design seismic level, small base-isolation bearings specifically designed for nuclear plant equipment were developed [13]. It should be noted that for export-standard nuclear power plants, the design-basis earthquake (DBE) is set at a safe shutdown earthquake (SSE) level of 0.3 g, and they are designed to have a seismic performance level of up to 0.5 g. However, in the event of a BDBE, some equipment vulnerable to earthquakes may be affected. To enhance the seismic performance of such earthquake-vulnerable equipment, research on the development of small lead–rubber bearings (LRB) for base-isolation devices was conducted.

When making small-sized LRBs, there are limitations in the manufacture of the stacked rubber in the baseplate, resulting in a lower shape factor. This leads to decreased stability compared with larger base-isolation devices for building structures, and the energy dissipation effect is somewhat insufficient. To address these issues and improve damping performance, lead cores are inserted at the center of the base-isolation devices to enhance their base-isolation performance.

In this study, the unit support weights of the small LRBs were initially manufactured with four different designs, 1-ton, 2-ton, 5-ton, and 10-ton LRBs, and static tests were conducted. Since much of the internal equipment in nuclear power plants is low-weight individual devices, the design was optimized primarily for 1-ton LRBs. The detailed design process is explained in reference [13]. Low-capacity base-isolation bearings are highly necessary for technical development and have high applicability at nuclear plant sites, especially for components like nuclear plant instrumentation. Furthermore, among design approaches with different sizes and dynamic characteristics, base isolation performance, including natural frequencies and shape factors, was compared and evaluated to determine the optimal design. The design details, including cross-sectional shape and the design specifications of the LRB finally chosen and used in this study, are provided in Figure 1 and Table 1, respectively.

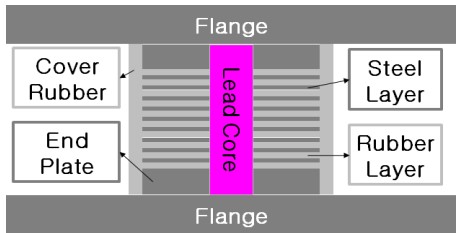

**Figure 1.** LRB cross-sectional shape.

**Table 1.** Design specifications for 1-ton LRB for NPP equipment.

| Properties | Values |
|---|---|
| Outer Diameter (mm) | 100 mm |
| Design Horizontal Freq. | 2.3 Hz |
| Shape Factors, S1/S2 | 9.9/5.0 |

Table 2 indicates the mechanical properties of the rubber and lead materials used in the fabrication of small equipment LRBs. The properties of the rubber material are very similar to natural rubber, and pure lead-like materials were used for the lead core [13].

**Table 2.** Mechanical properties of the 1-ton LRB.

| Materials | Properties | Values |
|---|---|---|
| Rubber | Shear Modulus (MPa) | 0.3 |
| | Young's Modulus (MPa) | 0.9 |
| | Bulk Modulus (GPa) | 1.96 |
| | Tensile strength (ksi) | 2.5~3.5 |
| | Strain (%) | 750~850 |
| | Density (g/cm$^3$) | 0.93 |
| Lead | Shear Modulus (MPa) | 8.33 |

*3.2. Seismic Response Sensitivity of Base-Isolated Structures by Mass Eccentricity*

In this paper, the response due to structural stiffness eccentricity is not considered. Therefore, a small base-isolation bearing designed for equipment, developed in previous studies, was applied, assuming a constant support stiffness. Consequently, a base-isolated cubic design with lumped mass was selected as the analysis model for the upper structure, which facilitates setting the mass eccentricity sensitivity of the seismic response.

To reduce the significant computational time required for seismic time history analysis when modeling small base isolators at full scale and performing nonlinear analysis, this study simplified the small base isolators by representing them as spring elements for sensitivity analysis.

For reference, Figure 2a depicts the configuration where the four 1-ton base isolators developed earlier are positioned at the corners of the 4-ton cubic mass model's underside for support. Figure 2b shows the simplification for analysis efficiency, where each base isolator is modeled with one vertical (Z) and two horizontal (X, Y) spring elements, totaling 12 spring elements. The stiffnesses of the springs in the horizontal and vertical directions are calculated as follows.

$$K_h = (G \times A) / T_R \tag{2}$$

$$K_v = (E_C \times A) / T_R \tag{3}$$

($K_h$: Horizontal stiffness, $K_v$: Vertical stiffness. $T_R$: The total thickness of the rubber layer in the base isolator, $G$: The Shear modulus of the rubber, $E_C$: The elastic modulus adjusted based on compression characteristics, $A$: Effective design area).

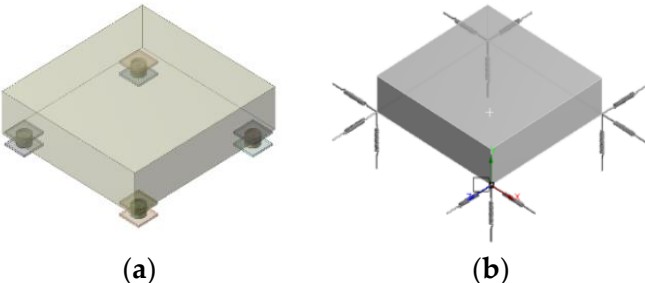

**Figure 2.** Sensitivity analysis models for a base-isolated structure. (**a**) Dummy mass on the LRB model; (**b**) Dummy mass on the equivalent spring model.

To investigate the changes in horizontal dynamic characteristics and response sensitivity due to the one-axis mass eccentricity of the structure, seismic analysis was conducted using code-based methods [14]. In the sensitivity analysis, as shown in Figure 3, the mass eccentricity was defined as 0% and 100% when the mass center of the upper structure (Dummy Mass) was positioned at the center and at the edge, respectively. In this context, the analysis was theoretically extended up to 100% eccentricity for the purpose of the study. However, in actual structural design, eccentricities exceeding 50% are very unusual.

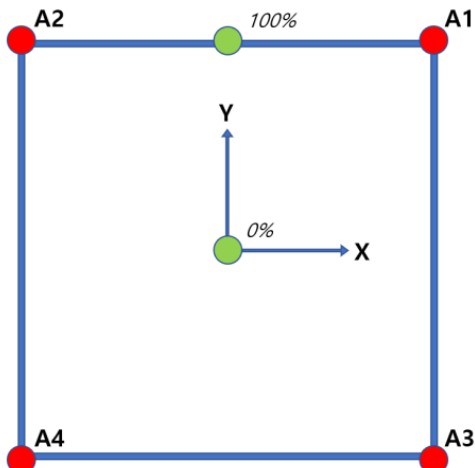

**Figure 3.** Definition of mass eccentricity value in Y direction and locations of A1 to A4 points.

It should be noted that the points from A1 to A4 in Figure 3 represent the corner positions of the upper structure, which are the actual locations of the acceleration sensors used in subsequent vibration testing, as described later. By varying the upper structure's mass eccentricity from 0% to 100% in 1% intervals, modal analysis and response spectrum analysis were conducted to observe the impact on displacement and acceleration seismic responses.

### 3.3. Results of the Sensitivity Analysis of Seismic Response by Mass Eccentricity

First, we aimed to examine the changes in the dynamic characteristics of the base-isolated structure as the eccentricity increased. Modal analysis results indicate that for 0% eccentricity (non-eccentric), the first, second, and fourth modes appear as translational (X, Y, Z) modes, the third mode as a horizontal torsional (rotation) mode, and the fifth and sixth modes as vertical rocking modes as shown in Table 3. As mass eccentricity increases, when observing changes in the dynamic characteristics of the base-isolated upper structure, torsional modes appear in lower-order modes, and, in some translational modes, a mixed mode with torsional behavior becomes more evident.

**Table 3.** Mode characteristics based on eccentricity in base-isolated structures.

| | 0% Eccentricity | | 100% Eccentricity | |
| --- | --- | --- | --- | --- |
| Mode | Hz | Mode Shapes | Hz | Mode Shapes |
| 1 | 2.3 | Translation (X) | 1.7 | Translation (X) + Rotation (RZ) |
| 2 | 2.3 | Translation (Y) | 2.3 | Translation (Y) |
| 3 | 3.3 | Rotation (RZ) | 4.6 | Rotation (RZ) |
| 4 | 36.9 | Translation (Z) | 22.7 | Translation (Z) + Rocking (RX) |
| 5 | 50.5 | Rocking (RY) | 50.5 | Rocking (RY) |
| 6 | 53.2 | Rocking (RX) | 86.6 | Rocking (RX) |

As the mass eccentricity increases, the contribution of the torsional mode increases, leading to a mixed behavior of translational and torsional modes in the structure [15]. To summarize, the changes in the natural frequencies of each mode as the horizontal single-axis mass eccentricity increases were observed and are presented in Figure 4. As the mass eccentricity of the base-isolated upper structure increases from 0% to 100%, the natural frequency of the first mode does not decrease much; however, the source of mode behavior changes from translation to a combination of translation and rotation. This is likely due to the torsional behavior caused by mass eccentricity. The second mode remains unaffected by horizontal direction eccentricity, as expected. In the case of the third mode, the frequency actually increases from 3.3 Hz to 4.6 Hz. This is attributed to an increase in the contribution of the first mode due to eccentricity, coupled with a decrease in horizontal torsion due to no eccentricity. The fifth mode, similar to the second mode, remains unaffected by eccentricity, resulting in no change in its natural frequency. The fourth and sixth modes exhibit characteristics similar to the changes observed in the first and third modes. Therefore, when the upper structure of the base-isolated structure experiences mass eccentricity, the torsional mode becomes the dominant mode, leading to changes in dynamic behavior with increased contributions to the response.

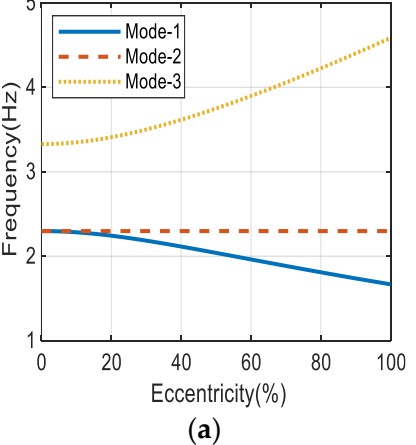 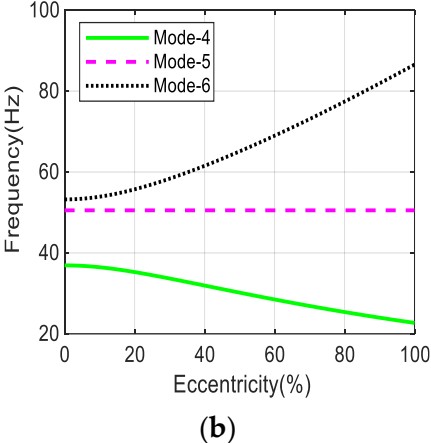

(**a**) (**b**)

**Figure 4.** Change in modal frequencies with eccentricity. (**a**) Frequencies for Mode-1 to Mode-3; (**b**) Frequencies for Mode-4 to Mode-6.

Next, to examine structural response characteristics for the input of seismic motion, 3D seismic response analysis was conducted using the design response spectrum (DRS) applied in domestic export-standard nuclear power plants, as shown in Figure 5. In Figure 5a, the horizontal direction (EW) DRS is compared with the test response spectrum (TRS) and the required response spectrum (RRS). In Figure 5b, the acceleration time history corresponding to the DRS is shown.

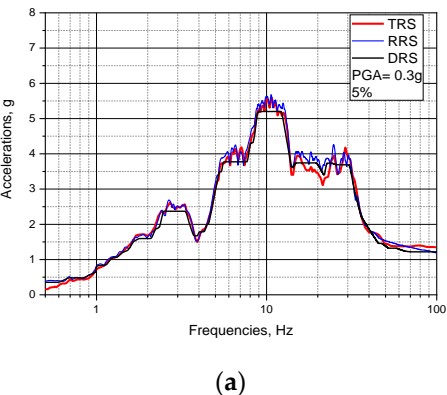
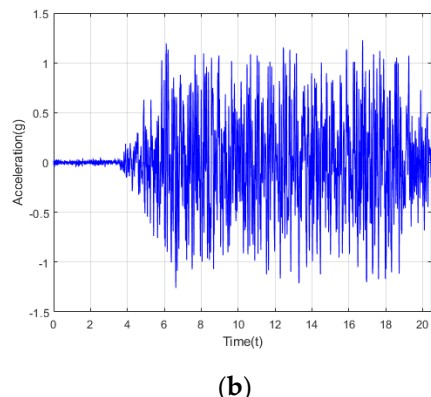

(**a**)                    (**b**)

**Figure 5.** Seismic inputs for analysis. (**a**) Acceleration response spectra; (**b**) Acceleration time histories for 0.3 g DRS.

Next, to observe the seismic response impact of the eccentricity in a base-isolated structure, the average acceleration response at the center of the upper structure is analysed and reviewed. Within an eccentricity of 10%, both the X and Y directions show minimal changes in acceleration. However, beyond 10% eccentricity, the average acceleration in the X direction starts to decrease significantly, reaching a minimum acceleration that is about 20% lower than the initial value at around 50% eccentricity. Beyond 50% eccentricity, the X-direction acceleration gradually increases again.

In contrast, the Y-direction average acceleration shows a linear response increase of up to about 15% as eccentricity increases. The eccentricity sensitivity of horizontal acceleration response at the center of gravity is shown in Figure 6.

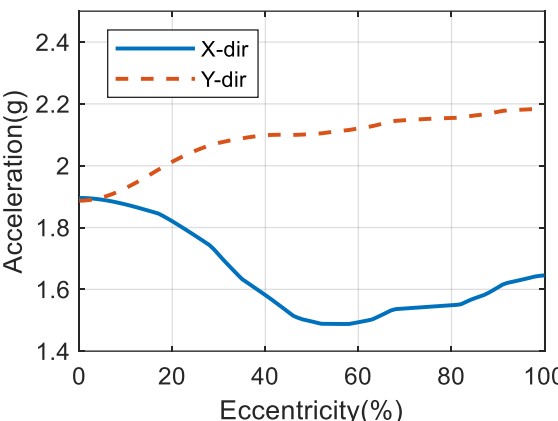

**Figure 6.** Eccentricity sensitivity of horizontal acceleration at the mass center.

In order to examine the changes in more detail, we checked the acceleration responses at the four corner points from A1 to A4, which correspond to the corner positions of the upper structure, as shown in Figure 3. Since A2 and A1 are symmetrical with respect to the Y axis, we omitted some of them and present them in Figure 7.

The acceleration response in the Y direction, which serves as the reference for mass eccentricity, exhibits the same response characteristics as the average acceleration in Figure 8. However, as seen in Figure 7a, as eccentricity increases, the X-direction acceleration at point A2, where the center of mass gets closer, initially increases due to the eccentricity of the mass, but then linearly decreases after about 20% eccentricity. This is because, as indicated by the earlier modal analysis results, as eccentricity increases, the contribution of the torsional mode rises, leading to an initial increase in acceleration due to the combined behavior of translation and torsion modes.

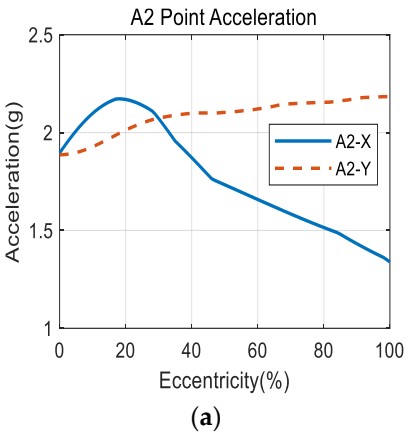
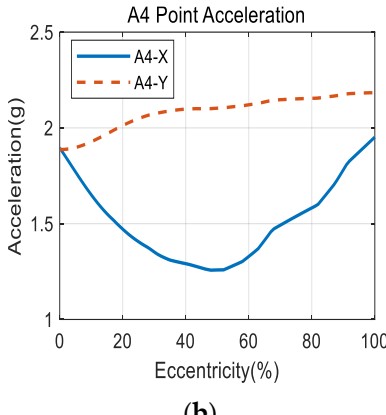

**Figure 7.** Eccentricity sensitivity of horizontal acceleration at A2 and A4 points. (**a**) A2 Point; (**b**) A4 Point.

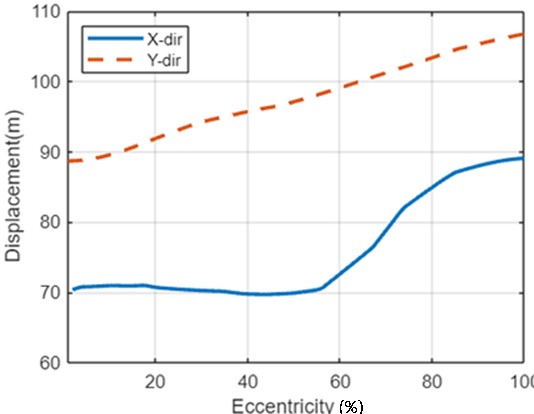

**Figure 8.** Eccentricity sensitivity of horizontal displacement at the mass center.

However, as shown in Figure 7b, as eccentricity increases, the change in the center of mass's position due to the change in the rotational center position results in variations in acceleration values at specific points. Due to this phenomenon, the X-direction acceleration responses at A2 and A4 exhibit opposite characteristics. Therefore, during the seismic design of structures with mass eccentricity, it is important to pay attention to such peak response characteristics at around 20% eccentricity depending on the upper structure's position.

Nextl, the displacement sensitivity to mass eccentricity during seismic responses is examined. As shown in Figure 8, the Y-directional displacement response exhibits a linear increase of up to 25% as eccentricity increases. In contrast, the X-directional displacement initially shows little change in response, but starting from around 50% eccentricity, it exhibits an approximately 30% increase. This indicates that special attention is required when dealing with significant eccentricities, particularly in the case of large eccentricities, in the context of X-direction displacement.

The sensitivity of displacement response for points A2 and A4, as shown in Figure 9, indicates that, as eccentricity increases, there is an increase in the influence of torsional behavior, leading to an increase in displacement response. In other words, as eccentricity increases, the response at point A2, where the mass center is closer, increases due to the influence of torsion behavior. Conversely, the response at point A4, which is farther from the mass center, decreases in overall response due to the out-of-phase effect of increased X-direction displacement. For instance, at eccentricity of around 20%, the maximum displacement response can be observed to increase by approximately 20% or decrease by nearly 30%, depending on the location of the upper structure.

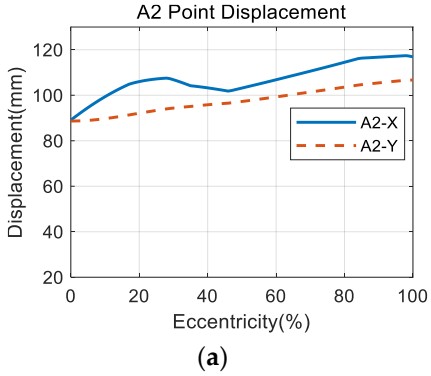 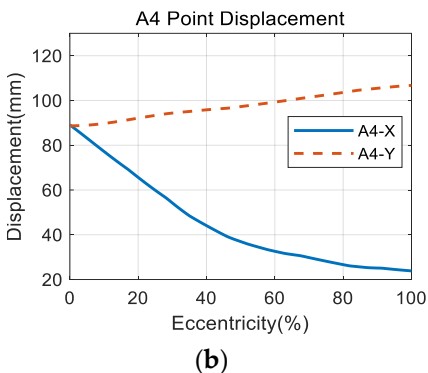

(**a**)              (**b**)

**Figure 9.** Eccentricity sensitivity of horizontal displacement at A2 and A4 points. (**a**) A2 Point; (**b**) A4 Point.

In summary, the analysis results for the response spectra based on mass eccentricity in the one-axis direction indicate that, as eccentricity increases, the combined behavior due to the increased contribution of torsion leads to varying acceleration or displacement responses at specific locations of the upper structure, depending on its position.

In the following analysis, a sensitivity analysis of seismic response due to 2D mass eccentricity in both horizontal directions was conducted. For the input of seismic motion, the EW direction design response spectrum from Figure 5 was utilized, along with the NS direction DRS, to perform acceleration and displacement response spectrum analysis.

Figure 10 illustrates the changes in acceleration response for the X direction and Y direction while simultaneously increasing mass eccentricity in both horizontal directions. In Figure 10a, it can be observed that, for X-direction acceleration, the response sensitivity due to 2D mass eccentricity, where both Y-direction and X-direction eccentricities exist, shows nonlinear variations. As seen in Figure 10b, a similar trend is observed for Y-direction acceleration. In summary, the common feature is that, while in some regions the response may decrease, when mass eccentricity increases simultaneously in both horizontal directions, acceleration response can significantly increase. Thus, caution is required.

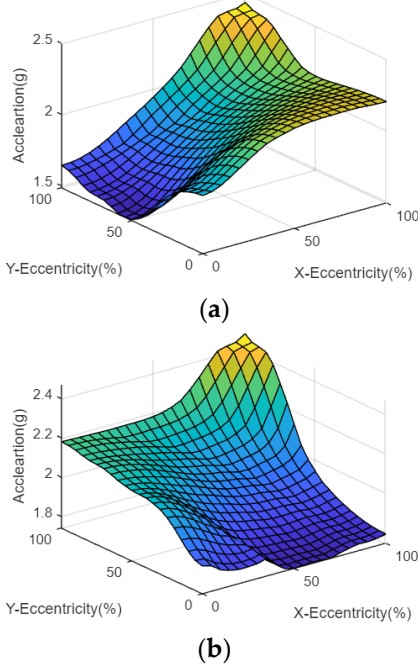

(**a**)

(**b**)

**Figure 10.** 2-D mass eccentricity effect on horizontal max. acceleration. (**a**) Effect on max. acceleration in the X direction; (**b**) Effect on max. acceleration in the Y direction.

Figure 11 is similar to Figure 10, showing the influence on displacement response for the X direction or Y direction while simultaneously increasing mass eccentricity in the horizontal direction. As shown in Figure 11a, in the case of X-direction displacement, when Y-direction mass eccentricity is low, the displacement response linearly increases in proportion to the eccentricity. For X-direction non-eccentric cases, there is little effect from Y-direction eccentricity on displacement response until the X-direction eccentricity reaches around 50%, at which point the sensitivity to eccentricity increases significantly.

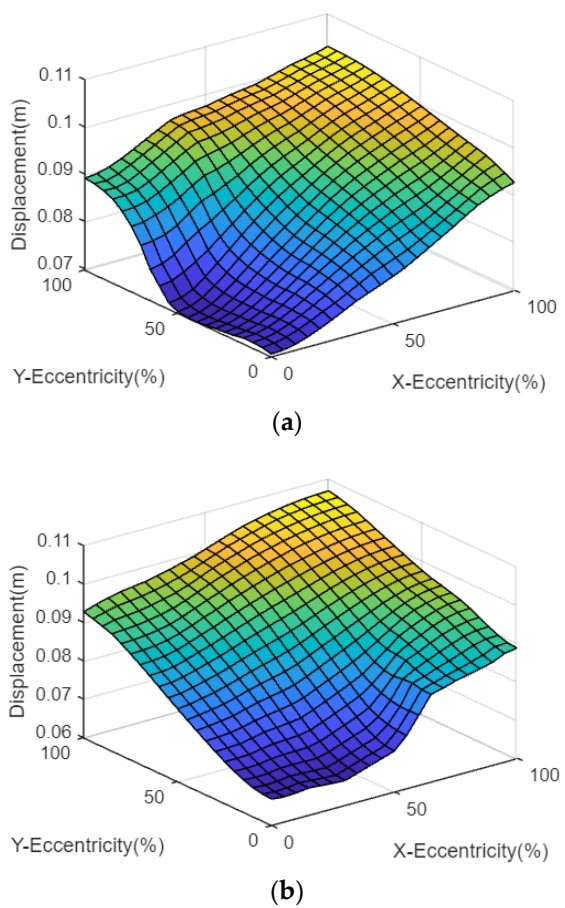

(**a**)

(**b**)

**Figure 11.** 2-D mass eccentricity effect on horizontal max. displacement. (**a**) Effect in the X direction; (**b**) Effect in the Y direction.

In Figure 11b, Y-direction displacement demonstrates a different behavior. When X-direction mass eccentricity is low, the response sensitivity to Y-direction mass eccentricity remains relatively constant. As both horizontal mass eccentricities increase, Y-directional maximum displacement can increase by approximately 60%.

## 4. Simulation Analysis for the Shaking Table Test

To investigate the effects of the mass eccentricity of the superstructure on the laminated elastomeric bearing (LRB), several cubic steel blocks easy to reform by assembly were fabricated. For the efficiency of the tests, the superstructure was designed for three different configurations: (1) non-eccentric (Non-Ecc, or Con), (2) eccentricity of 12.5% (Ecc1), and (3) Eccentricity of 25% (Ecc2). These configurations were designed to facilitate model changes for on-site reassembly, as shown in Figure 12. The total mass of each model was set to 4 tons, considering support by four 1-ton capacity laminated elastomeric bearings. The actual eccentricities for each model can be calculated as shown in Table 4. Prior to conducting the three-axis shake table tests for these models, seismic response analyses were carried out to simulate the tests.

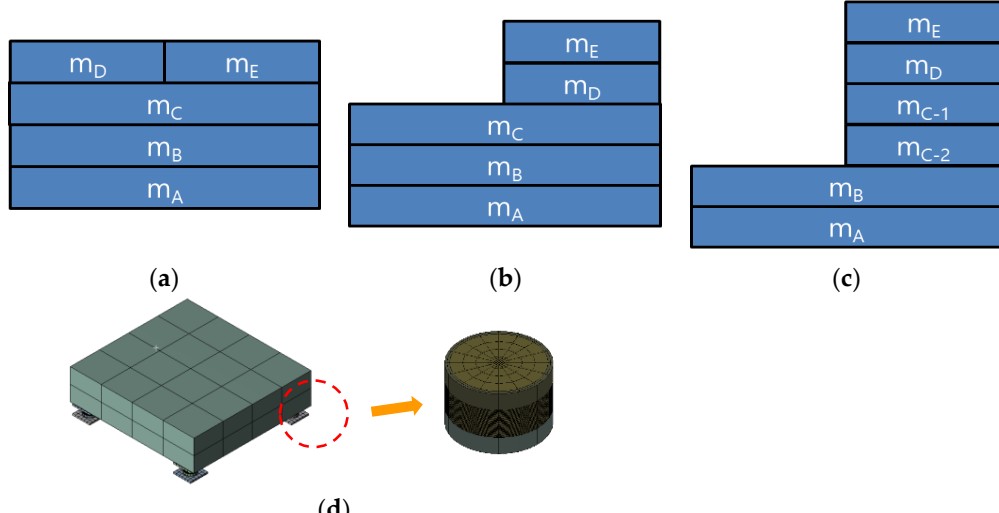

**(a)**          **(b)**          **(c)**

**(d)**

**Figure 12.** Model shapes for simulation analysis for the shaking table test. (**a**) Non-eccentric mass; (**b**) Eccentric mass 1 (Ecc1); (**c**) Eccentric mass 2 (Ecc2); (**d**) FEM model for non-eccentric mass for example.

**Table 4.** Mass center location and eccentricity for each model.

| Model Name | Location [mm] | Eccentricity [%] |
|---|---|---|
| Non-Ecc. (Con) | 0 | 0 |
| Ecc1 | 77 | 12.5 |
| Ecc2 | 154 | 25 |

To predict the three-axis shaking table test results, FEM models were created for each test specimen, and seismic analyses were conducted. The same seismic input data from Figure 5 were used as the input for ground motion. To perform response spectrum analysis of the FEM models, the maximum acceleration responses at four corner locations where accelerometers were installed (A2 to A5) were reviewed.

Tables 5 and 6 show a comparison of the results for maximum acceleration and displacement responses between sensitivity analysis (SA) and FEM simulation analyses for the test, respectively.

**Table 5.** Max. acceleration (g) b/w sensitivity and simulation analyses.

| Mass Eccentricity | Base-Fixed | Base-Isolated | |
|---|---|---|---|
| | | Sensitivity Anal. | Simulation Anal. |
| Non-Ecc. (0%) | 5.25 | 1.89 | 1.89 |
| Ecc1 (12.5%) | 5.06 | 1.86 | 1.87 |
| Ecc2 (25%) | 4.65 | 1.78 | 1.75 |

**Table 6.** Max. displacement comparison by mass eccentricity in analyses for base conditions.

| Mass Eccentricity | Base-Fixed | Base-Isolated | |
|---|---|---|---|
| | | Sensitivity Anal. | Simulation Anal. |
| Non-Ecc. (0%) | 13 cm | 88 | 88 |
| Ecc1 (12.5%) | 13 cm | 55 | 67 |
| Ecc2 (25%) | 12 cm | 82 | 87 |

In this comparison, the maximum seismic response differences between the sensitivity analysis and simulation analysis, especially for the mass eccentricities of the base-isolated upper structure, were 0% in the non-eccentric condition, 12.5%, and 25%, respectively.

As shown in the tables, the maximum acceleration responses of the three different eccentricity models from both analyses show mostly good matches without remarkable differences. However, the maximum displacements were larger in the FEM simulation analysis compared with the sensitivity analysis. This discrepancy is attributed to the rocking behavior induced by the vertical mass eccentricity of the simulation model, which is not considered in sensitivity analysis. In summary, while the acceleration responses were relatively consistent, the maximum displacements were influenced by eccentricity-induced rocking behavior in the simulation analysis compared with the sensitivity analysis.

## 5. Shaking Table Test for Base-Isolated Structures with Mass Eccentricity

To assess the seismic response impact caused by the mass eccentricity of the base-isolated upper structure, sensitivity analysis and simulation analysis were conducted, as explained earlier.

Subsequently, to validate the analysis results obtained, actual tests were carried out using a three-axis vibration table. It is worth noting that the vibration table used in this test had a capacity of 30 tons for dynamic experiments in three directions, and its detailed specifications can be found in Table 7. The test models consisted of the upper structure supported by four small LRBs, as previously described. Specifically, three models were used, each with different mass eccentricities, as shown in Figure 13. Additionally, to investigate the impact on the secondary structure, three different substructures, designed to have natural frequencies of 5 Hz, 10 Hz, and 30 Hz, were attached to the test table and upper structure, respectively, for response measurements [16].

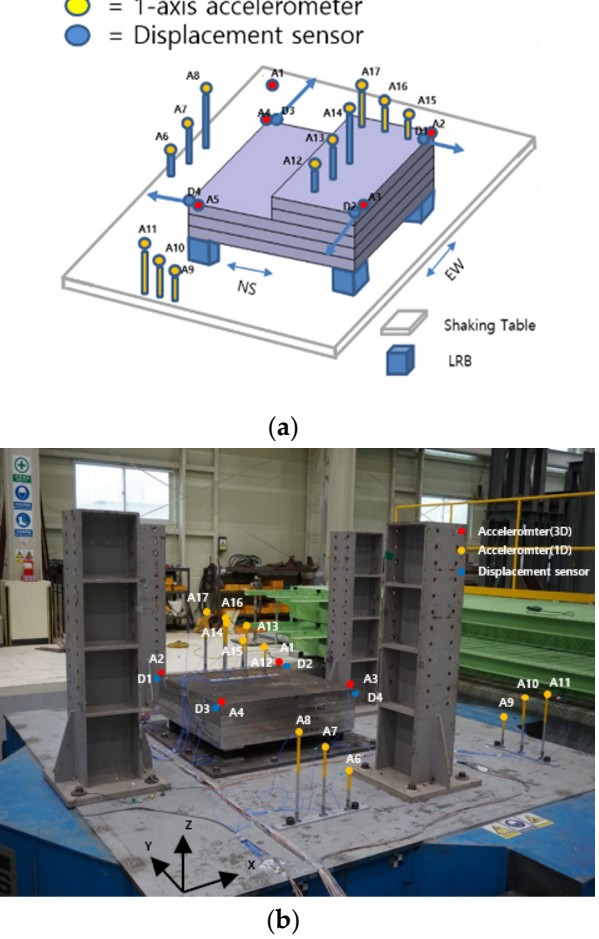

(**a**)

(**b**)

**Figure 13.** Testing of mass eccentric structures using a 3-D table. (**a**) Schematic of sensor locations on the Ecc1 structure; (**b**) Preparation configuration for the 3-D table test.

**Table 7.** Technical specification for the 3-D shaking table.

| Technical Variables | Values | | |
|---|---|---|---|
| Size [m] | 4.0 × 4.0 | | |
| Max Loading (Force/Moment) | 300 kN/1200 kN·m | | |
| Frequency Range (Hz) | 0.1–60 | | |
| Control Axes | 6 DOF | | |
| Acceleration at Full Payload (g) | X | Y | Z |
| | 1.2 | 1.2 | 0.8 |
| Maximum Stroke (mm) | X | Y | Z |
| | ±300 | ±200 | ±150 |

Figure 13a shows the sensor locations and beam models for the test of the eccentric mass 1 (Ecc1) structure as an example, and Figure 13b shows the picture of the actual test setup. Two sets of three different beams were installed both on the shaking table and also on the top of the eccentric mass structures to investigate the mass eccentricity effect of the upper structure on the seismic response of the secondary structure. This setup was designed to assess the response impact on the secondary structure in both the X and Y directions.

When actual vibration tests were conducted using the seismic input data from Figure 5, it was observed that the acceleration response of the base-isolated superstructure decreased in the frequency range of interest around 5 Hz to 50 Hz, as shown in Figure 14 [16].

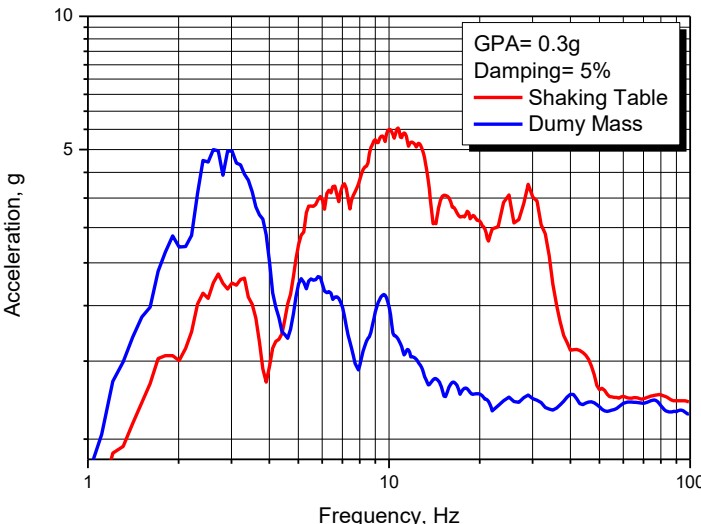

**Figure 14.** Effect of base isolation on acceleration responses (0.3 g, EW).

Specifically, a peak occurred around the frequency of the small base isolator's isolation frequency, approximately near 2.3 Hz, as expected. The response gradually decreased up to around 20 Hz and then maintained a relatively constant value for the remaining frequency range.

As shown in Figure 15, it can be observed that, for the Ecc1 and Ecc2 models (e.g., with mass eccentricities of 12.5% and 25%, respectively), the acceleration responses of the upper structure increased in the overall frequency range in comparison with non-eccentric models. In the overall frequency range, increasing eccentricity leads to a more than 40% increase in acceleration response compared with the non-eccentric case. However, in the frequency range around 3 Hz and 4 Hz, as eccentricity increases, acceleration response can increase by up to approximately 100%.

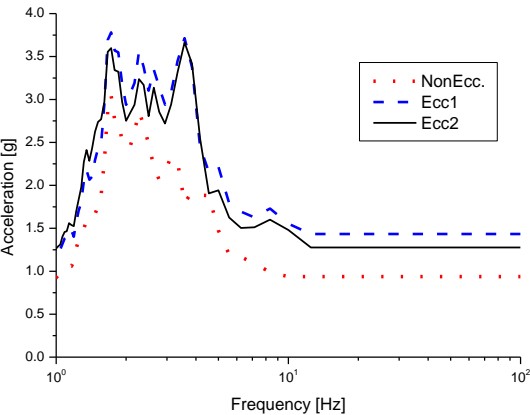

**Figure 15.** Acceleration responses of base-isolated structures by mass eccentricity.

This behavior is attributed to the combined effects of eccentricity-induced torsional behavior and the fact that the earthquake excitation input has the highest energy content at around 10 Hz. This results in a significant increase in acceleration response in the range of 4 Hz to 15 Hz for the Ecc2 case.

Table 8 compares the seismic acceleration responses of the base-isolated upper structure (dummy mass) for three models: non-eccentric (Con), eccentricity of 12.5% (Ecc1), and eccentricity of 25% (Ecc2). The comparison includes sensitivity analysis (SA), finite element model (FEM) analysis, and the seismic response in the three-axis vibration table test model. The measurement locations are the four corners (A2–A5) and the center point of the upper structure, and the results are given in terms of acceleration responses.

**Table 8.** Analysis and test results for mass-eccentric base-isolated structures.

| Loc | Non-Ecc (0%) | | | Ecc1 (12.5%) | | | Ecc2 (25%) | | |
| --- | --- | --- | --- | --- | --- | --- | --- | --- | --- |
| | SA | FEM | Test | SA | FEM | Test | SA | FEM | Test |
| A2 | 1.89 | 1.91 | 2.15 | 2.12 | 2.03 | 1.63 | 2.14 | 1.92 | 1.73 |
| A3 | 1.89 | 1.91 | 1.53 | 2.12 | 2.03 | 1.56 | 2.14 | 1.93 | 1.72 |
| A4 | 1.89 | 1.89 | 1.81 | 1.60 | 1.52 | 1.09 | 1.41 | 1.36 | 1.01 |
| A5 | 1.89 | 1.89 | 1.69 | 1.60 | 1.52 | 2.04 | 1.41 | 1.38 | 1.12 |
| Center | 1.89 | 1.89 | 1.80 | 1.86 | 1.78 | 1.58 | 1.78 | 1.65 | 1.40 |

This table reveals that, for the base-isolated structure, as the eccentricity of the upper structure's mass increases, the responses at individual corners may exhibit irregular behavior, but the response at the center tends to decrease. This behavior is likely due to increased torsional behavior with increasing eccentricity, which can offset translational motion. While the responses at each corner of the base-isolated structure show relatively minor differences between the sensitivity analysis and the finite element model analysis, the response of the test model shows differences of more than 30% in some locations compared with the analysis results. This difference is attributed to the fact that, in the sensitivity and FEM analyses, only horizontal mass eccentricity was considered, while in actual tests, both horizontal and vertical mass eccentricities were affected. Therefore, a combination of horizontal rotation and vertical rocking behavior could have lead to variations in the vertical direction response [17].

Figure 16 shows graphs illustrating the acceleration-response spectra of a beam-shaped secondary structure installed on top of the base-isolated upper structure and the shaking table. Additionally, the responses of beams installed on top of the base-isolated non-eccentric and eccentric upper structures are also compared. So to speak, the figures explain the effects of base isolation and the mass eccentricity of the upper structure on secondary structure response through acceleration-response spectra.

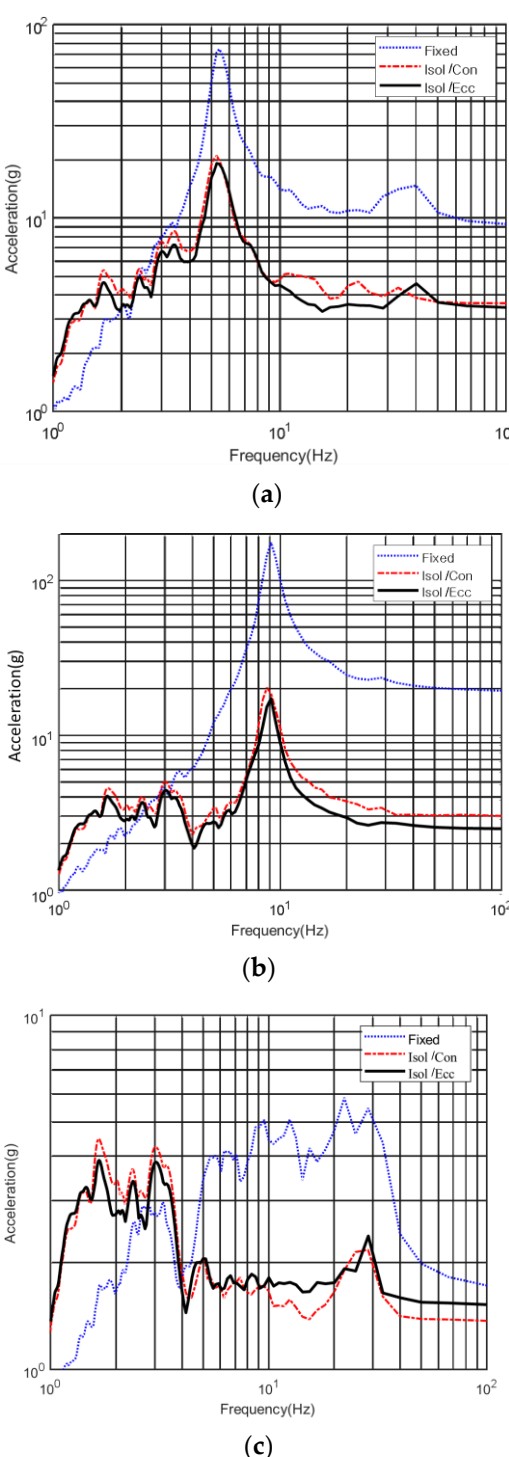

**Figure 16.** Comparison of mass eccentricity effects in secondary structures. (**a**) Response spectra for the 5 Hz beam; (**b**) Response spectra for the 10 Hz beam; (**c**) Response spectra for the 30 Hz beam.

In the case of the beam structure with a natural frequency of 5 Hz, as expected, there is a peak around 5 Hz, and the response is significantly higher in the vicinity of the base isolation frequencies (1~3.5 Hz) as shown in Figure 16a. However, overall, the response decreases in other frequency ranges, indicating that the base isolation effect is well maintained even for the secondary structure. For the beam structure with a natural frequency of 10 Hz, a peak occurs around 10 Hz, similar to the characteristics of the 5 Hz beam. Except for frequencies below 3.5 Hz, the response is lower compared with non-base-isolated structures. The reduction in response amplitude at the peak is 20% higher

for the 10 Hz beam, and this is attributed to the seismic energy input being significant at around 10 Hz.

Figure 16a,b demonstrates that the acceleration response of the secondary structure with natural frequencies of 5 Hz and 10 Hz, installed on top of the eccentrically base-isolated structure, shows minimal influence from eccentricity. These differences are not considered to be very significant due to mass eccentricity across various frequency ranges. However, the mass eccentricity effect of the primary structure on the secondary structure can increase by around 30–40% in the frequency range above 10 Hz, and this should be considered in the design. The base isolation effect is preserved effectively. However, as shown in Figure 16c, for the 30 Hz secondary structure, the acceleration response mainly increases above 4 Hz due to the mass eccentricity of the base-isolated upper structure. Therefore, in the case of a secondary structure with high stiffness and high natural frequencies, the base isolation effect may be reduced due to the mass eccentricity of the upper structure. It is essential to be cautious in such situations.

## 6. Conclusions

In this study, an investigation was carried out to assess the effects of seismic responses when mass eccentricity occurs in the base-isolated concentrated mass structure within a nuclear power plant. Sensitivity analysis, simulation analysis, and vibration table tests were employed for this purpose. Generally, when mass eccentricity occurs in the base-isolated upper structure, the dynamic characteristics are influenced by torsional or rocking behavior. The summarized findings are as follows:

1. In case of horizontally mass-eccentric base-isolated structures, torsional modes can become dominant modes, and, as eccentricity increases, the contribution of torsional response increases.
2. For base-isolated upper structures, the influence of eccentric mass on the seismic response may not be significant compared with non-base-isolated structures. However, seismic response can increase or decrease by more than 40% depending on the location in the structure, so caution is required during design.
3. Actual tests using a 3D shaking table showed similar trends to sensitivity analysis and simulation results for horizontal motion.
4. The seismic response of secondary structures installed in base-isolated upper structures is relatively less affected by the mass eccentricity of the upper structure; however, it can reach around a 30–40% increase in the frequency range above 10 Hz.

These findings provide valuable insights into the seismic response of base-isolated structures with irregular mass distribution and emphasize the importance of considering horizontal and vertical mass eccentricity effects during design and analysis.

**Author Contributions:** Conceptualization, T.-M.S.; methodology, T.-M.S. and B.-C.L.; validation, T.-M.S.; formal analysis, T.-M.S.; investigation, T.-M.S. and B.-C.L.; writing—original draft preparation, T.-M.S.; writing—review and editing, T.-M.S. and B.-C.L.; funding acquisition, T.-M.S. All authors have read and agreed to the published version of the manuscript.

**Funding:** This study was funded by the Ministry of Trade, Industry, and Energy through KETEP. (Korea Institute of Energy Technology Evaluation Planning) (No. 20224B10200080).

**Institutional Review Board Statement:** The study was conducted according to the guidelines of the Declaration of Helsinki, and approved by the Institutional Review Board.

**Informed Consent Statement:** Informed consent was obtained from all subjects involved in the study.

**Data Availability Statement:** The data presented in this study are available on request from the corresponding author. The data are not publicly available due to preventing misuse for unwanted purpose by third party.

**Acknowledgments:** This paper was supported by the Korea Institute of Energy Technology Evaluation and Planning (KETEP) under Grant No. 20224B10200080.

**Conflicts of Interest:** The authors declare no conflict of interest.

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
