# Peer review of "Seismic Response Effect on Base-Isolated Rigid Structures by Mass Eccentricity in Nuclear Plants"

_applsci, doi:10.3390/app132413330_

Round 1
Reviewer 1 Report
Comments and Suggestions for Authors
Please check the attachment.

Minor editing of English language required
Author Response
Reviewer #1
The effects of mass eccentricity on the seismic responses of nuclear power facilities were investigated experimentally, theoretically, and numerically. Mass eccentricity may amplify seismic responses, thereby endangering the security of nuclear power facilities. Therefore, it is worthwhile to investigate this phenomenon. Please find the detailed comments as follows:
- The study conducted in this paper an outstanding investigation into the impact of eccentric mass on the seismic responses of nuclear power facilities. What steps might be taken to prepare a nuclear power plant for this consequence? Certain SMA devices and SMA-based isolators are capable of constraining the excessive displacement of isolated structures.
For instance, Choi, E.; Lee, D. H.; Choei, N. Y., Shape memory alloy bending bars as seismic restrainers for bridges in seismic areas. International Journal of Steel Structures 2009, 9 (4), 261-273. Deng, J. D.; Hu, F. L.; Ozbulut, O. E.; Cao, S. S., Verification of multi-level SMA/lead rubber bearing isolation system for seismic protection of bridges. Soil Dynamics and Earthquake Engineering 2022, 161. Could you provide a discussion of the potential use of these technologies in isolated nuclear power plants?
Author’s Notes 1;
SMA can be a good option for the purpose, and some studies have been tried for the materials of the nuclear fuel components. However, there are a couple of processes required for convincing the regulatory body in practice, as I know. And the issue is a little out of the main point of this paper. Many thanks for the concern about seismic enhancement of NPP!
- When discuss on Table 8, it stated that ‘…the response of the test model shows differences of more than 30% in some locations compared to the analysis results. This difference is attributed to the fact that in the sensitivity and FEM analyses, only horizontal mass eccentricity was considered, while in actual tests, both horizontal and vertical mass eccentricities affected.’ Is there any data that can support this discussion? Is it possible that the discrepancy is attributable to variations in damping ratios between the simulation and experimental tests?
Author’s Notes 2;
OK! It is good comment. I agree that it is attributed to the variations in damping ratios between the simulation and experimental tests. The damping in analysis is mostly assumed to be higher for conservatism especially in NPP system design. However, the mass eccentricity of the 3 models in vertical direction are, 0, 12.5%, 25%, respectively in direct calculation which could affect the rocking response like the response trend by horizontal mass eccentricity. Thanks!
- It stated that ‘In summary, Figure 16(a) and Figure 16(b) demonstrate that the acceleration response of the secondary structure with natural frequencies of 5 Hz and 10 Hz, installed on top of the eccentrically base isolated structure, shows minimal influence from eccentricity. Although some differences are observed in response due to mass eccentricity across various frequency ranges, these differences are not considered significant or meaningful. The base isolation effect is preserved effectively.’ on lines 479-484. Where are the locations of these supporting structures? Given their position at the center of the artificial mass, it is unsurprising that their reactions would remain unchanged. 3.
Author’s Notes 3;
OK! It may be unsurprising that their reactions would remain unchanged. The result is shown only to recommend operating NPP that base-isolation of the equipment with mass eccentricity would be a good option not only for reducing the response but to solve the eccentric problem. In addition, there are not many studies done so far for the response effect or sensitivity of secondary structure by the mass eccentricity of base-isolated primary structure.
- Are the responses depicted in Figures 10 and 11 identical in the X and Y directions? Displaying the response in only one direction may be sufficient.
Author’s Notes 4;
OK! It will be modified to the comment.
- Could the specific locations of the three beams depicted in Figure 16 be specified on Figure 13? 5.
Author’s Notes 5;
OK! The locations for 5Hz, 10Hz, 30Hz beams are A14, A13, A12 in Figure.13, respectively. And the explanation is added in short to the paper.
- Do you derive the conclusion to the statement on lines 121-124 from any relevant literature? Please cite the relevant literature.
Author’s Notes 6;
OK! The relevant references and referred number to be added.
- It said that ‘For base-isolated upper structures, the influence of eccentric mass may not be significant compared to non-base-isolated structures.’ In conclusion 2. Please clarify the influence of eccentric mass on what?
Author’s Notes 7;
OK! It will be modified from “influence of eccentric mass” to “influence of eccentric mass on the maximum seismic response”.
- There is an over number of keywords. Please decrease the quantity to fewer than five.
Author’s Notes 8;
OK! It will be reduced to four such as mass eccentricity; seismic response; base-isolation; beyond design basis earthquake.
- Please assign numerical labels to the equations..
Author’s Notes 9;
OK! It will be added.
- The units assigned to the axes in Figure 8 are inaccurate. Please correct. There are some Korean characters On Figure 13a. Please change them to English. Please change
Author’s Notes 10;
Good comments! Correction will be made.
- Please rewrite ‘…the response effects of seismic responses…’.
Author’s Notes 11;
Good comments! Correction will be made.
Thanks for smart and good comments!

Reviewer 2 Report
Comments and Suggestions for Authors
Paper deals with the effect of equipment eccentricity in seismic-isolated nuclear power plant. In the paper numerical and experimental results are shown and commented. It is opinion of the Reviewer that the paper is well structures and report some interesting results. However, before the publication, the following comments should be taken into account.
Comment 1
In the introduction the novelty of the paper should be clearly stated by the Authors. Please, comment on this.
Comment 2
Section 2. Authors states that ‘… when center of gravity in the horizontal direction is eccentric, torsional effects occur along with bending, leading to reduced lateral load-bearing capacity’. This in some case may be also more pronounced if the structure is irregular along the height. Among the others, Authors may mention the following works:
· D’Amato, M., Gigliotti, R., Laguardia, R., 2019. Seismic Isolation for Protecting Historical Buildings: A Case Study. Frontiers in Built Environment 5:87. https://doi.org/10.3389/fbuil.2019.00087.
· Tena-Colunga, A., & Escamilla-Cruz, J. L. (2007). Torsional amplifications in asymmetric base-isolated structures. Engineering structures, 29(2), 237-247.
Comment 3
In the paper there is no indication about the numerical models implemented. Please, clarify on this.
Comment 4
Reviewer does not understand which the difference between ‘FEM analysis’ and ‘Numerical analyses’ within the paper. Sometime these two definitions are interchangeable and confusing. Please clarify on this.
Comment 5
Line 486-489. Please better clarify this statement.
Comment 6.
Conclusion should report only the main outcomes of the work presented. Please revise it.
Author Response
Reviewer #2
Paper deals with the effect of equipment eccentricity in seismic-isolated nuclear power plant. In the paper numerical and experimental results are shown and commented. It is opinion of the Reviewer that the paper is well structures and report some interesting results. However, before the publication, the following comments should be taken into account.
Comment #1
In the introduction, the novelty of the paper should be clearly stated by the Authors. Please, comment on this.
Author’s Notes 1;
OK! Following sentences to be added in the introduction such as,
“There have been many researches introducing the seismic response effect and warning about response increase of building structures by mass or stiffness eccentricity. However, there are not enough research about the effect of mass eccentricity on the seismic response of the base-isolated equipment as a secondary structure in primary building or operating plants.”
Comment #2
Section 2. Authors states that ‘… when center of gravity in the horizontal direction is eccentric, torsional effects occur along with bending, leading to reduced lateral load-bearing capacity’. This in some case may be also more pronounced if the structure is irregular along the height. Among the others, Authors may mention the following works:
- D’Amato, M., Gigliotti, R., Laguardia, R., 2019. Seismic Isolation for Protecting Historical Buildings: A Case Study. Frontiers in Built Environment 5:87. https://doi.org/10.3389/fbuil.2019.00087.
- Tena-Colunga, A., & Escamilla-Cruz, J. L. (2007). Torsional amplifications in asymmetric base-isolated structures. Engineering structures, 29(2), 237-247.
Author’s Notes 2;
OK! Recommended corrections are made as references with contents about referred papers such as, “It becomes more distinct if the structure is irregular along the height according to some researches [2,3].“ on lines 64, 65.
Comment #3
In the paper there is no indication about the numerical models implemented. Please, clarify on this.
Author’s Notes 3;
Yes, one of the FEM models added in Figure 12
Comment #4
Reviewer does not understand which the difference between ‘FEM analysis’ and ‘Numerical analyses’ within the paper. Sometime these two definitions are interchangeable and confusing. Please clarify on this.
Author’s Notes 4;
OK, agreed. Some expressions are modified for clarity such as,
1) From “two efficient mass.” to “three efficient mass.”, and from “Initial numerical simulation.” to “Simulation analyses using finite element method (FEM) models.” in abstract.
2) “initial finite element analysis (FEM) simulations.” -> “simulation analyses using the test models.” on line 54.
Comment #5
Line 486-489. Please better clarify this statement.
Author’s Notes 5;
Yes, it should be corrected. The sentence will be modified such as “shows minimal influence from eccentricity. These differences are not considered to be very significant due to mass eccentricity across various frequency ranges. However, mass eccentric effect of primary structure on the secondary structure can increase about 30-40% in concerning frequency range above 10Hz, and it should be considered in the design.”.
Comment #6.
Conclusion should report only the main outcomes of the work presented. Please revise it.
Author’s Notes 6;
Yes, the conclusions 3 and 4 to be modified such as,
- Actual tests using a 3-axis vibration table showed similar trends to sensitivity analysis and simulation results in horizontal motion.
- The seismic response of secondary structures installed in base-isolated upper structures is relatively less affected by the mass eccentricity of the upper structure, however, it can reach about 30%-40% increase in frequency range above 10Hz.
Thanks for smart and good comments!

Round 2
Reviewer 2 Report
Comments and Suggestions for Authors
Really interesting paper.
It may be published in the current revised version.